# Notch Signaling: An Emerging Paradigm in the Pathogenesis of Reproductive Disorders and Diverse Pathological Conditions

**DOI:** 10.3390/ijms25105423

**Published:** 2024-05-16

**Authors:** Sreesada Parambath, Nikhil Raj Selvraj, Parvathy Venugopal, Rajaguru Aradhya

**Affiliations:** School of Biotechnology, Amrita Vishwa Vidyapeetham, Kollam 690525, Kerala, India; sreesadar@gmail.com (S.P.); nikhilraj.ceib@gmail.com (N.R.S.); parvathy110883@gmail.com (P.V.)

**Keywords:** Notch, endometrium, Notch signaling, infertility, EMT, polarity, *D. melanogaster*

## Abstract

The highly conserved Notch pathway, a pillar of juxtacrine signaling, orchestrates intricate intercellular communication, governing diverse developmental and homeostatic processes through a tightly regulated cascade of proteolytic cleavages. This pathway, culminating in the migration of the Notch intracellular domain (NICD) to the nucleus and the subsequent activation of downstream target genes, exerts a profound influence on a plethora of molecular processes, including cell cycle progression, lineage specification, cell–cell adhesion, and fate determination. Accumulating evidence underscores the pivotal role of Notch dysregulation, encompassing both gain and loss-of-function mutations, in the pathogenesis of numerous human diseases. This review delves deep into the multifaceted roles of Notch signaling in cellular dynamics, encompassing proliferation, differentiation, polarity maintenance, epithelial–mesenchymal transition (EMT), tissue regeneration/remodeling, and its intricate interplay with other signaling pathways. We then focus on the emerging landscape of Notch aberrations in gynecological pathologies predisposing individuals to infertility. By highlighting the exquisite conservation of Notch signaling in Drosophila and its power as a model organism, we pave the way for further dissection of disease mechanisms and potential therapeutic interventions through targeted modulation of this master regulatory pathway.

## 1. Introduction

Since its discovery at the beginning of the 20th century, the Notch signaling pathway has been demonstrated as one of the main cell–cell communication channels across metazoans. Among its main functions, Notch signaling manifests a critical role in maintaining the polarity of the cell, in cell proliferation, differentiation, apoptosis, EMT (epithelial–mesenchymal transition), adhesion, invasion, and in deciding the cell fate [1,2]. It is also involved in remodeling and regeneration of tissues, and complex interaction with other signaling pathways.

Not surprisingly, the dysregulation of this pathway has been implicated in a variety of medical domains, demonstrating its importance in conditions like cancer [3], heart disease [4], neurological disorders [5], and gynecological illnesses. Notch signaling abnormalities are seen in a variety of cancers such as leukemia and solid tumors and thus present a possible target for therapeutic intervention. On the other hand, Notch signaling affects cardiogenesis and vascular maintenance in cardiovascular illnesses; disturbances to these processes result in vascular pathologies and congenital heart abnormalities. Furthermore, Notch signaling affects essential cellular processes in gynecological disorders, suggesting its potential as a therapeutic target to improve patient outcomes.

The unassuming *Drosophila melanogaster*, often relegated to as a fruit fly, has become vital in understanding the Notch signaling pathway. The Notch pathway’s remarkable conservation between Drosophila and humans has unraveled complexities in human diseases. Pioneer studies in Drosophila showed its crucial roles in the implication of this intricate network in orchestrating development, differentiation, and tissue homeostasis and provided some insights into pathologies upon Notch malfunction [6]. Drosophila’s rapid life cycle and genetic tools like CRISPR-Cas9 enable researchers to conduct intricate experiments with unparalleled precision [7,8]. By mimicking human disease-causing mutations, Drosophila researchers uncover how Notch dysregulation leads to various pathologies. Using Drosophila models, scientists decode the “Notch code”, shedding light on disease mechanisms and potential therapeutic interventions [9].

### Deciphering the Machinery: Understanding Notch Signaling Components and Organization

Notch was first discovered in a study of *Drosophila melanogaster* mutants with notched wings in the 1910s [10]. The gene’s structure allows for the encoding of a single large protein by a significant 10.4 kb RNA that contains characteristic signal and transmembrane sequences. The Notch protein is a transmembrane protein that has a unique extracellular domain with 36 repeating units that exhibit significant similarities to a number of mammalian clotting and growth factors [11]. When the specific ligand from adjacent cells binds to the Notch receptor, the receptor becomes activated. It results in two-step enzyme-mediated proteolytic cleavage of the Notch receptors. The first is S2 cleavage of the extracellular domain that is mediated by A Disintegrin and metalloproteinase (ADAM) secretase [12,13], and the second is S3 cleavage which is mediated by γ-secretase in the intracellular region of the receptor that results in the release of the Notch intracellular domain (NICD). The NICD then migrates to the nucleus where it interact with the DNA-binding domain, CBF1, Suppressor of Hairless, Lag-1 (CSL) in the case of mammals, and with other co-activators to activate the downstream Notch target genes [12,13].

Although the overall structure of Notch signaling is highly conserved across species, there are a few key differences between the Notch signaling system in Drosophila and mammals. A single Notch receptor in Drosophila is activated by the ligands Delta and Serrate after being broken down by the protease Kuzbanian [14]. In contrast, mammals have four different notch receptors (Notch 1–4) and five distinct ligands, including Jagged1, Jagged2, and Delta-like ligands (DLL1, DLL3, DLL4) which are all processed by the ADAM family of proteases [15] (Figure 1).

Notch signaling is also known to interact with other signaling pathways, which affects downstream targets and controls cellular functions. At various levels of the signal transduction cascade, the Notch and Wnt signaling pathways interconnect with one another either cooperatively or antagonistically. Numerous studies have demonstrated that Notch’s carcinogenic function involves communication with the Wnt signaling system through direct or indirect effects on Wnt pathway elements [16]. The Notch pathway also regulates Hedgehog signaling through modulation of the expression of its ligands and receptors [17,18]. Furthermore, Notch coordinates with the TGFβ signaling pathway for cell fate decisions [14,19].

There is also a crosstalk between Notch with other signaling pathways like insulin/TOR and myc during Drosophila AMP (adult muscle precursor) proliferation [13]. The crosstalk between Notch and other signaling pathways seems to be a conserved mechanism as in myogenic or osteoblast differentiation models, Notch coordinates this process with the BMP signaling pathway [20].

Such cooperative environment also occurs in the female reproductive system, as the human endometrium presents a multifaceted Notch signaling environment, encompassing various family members [21]. In this tissue, Notch interacts with the Wnt signaling pathway to regulate the activity of endometrial mesenchymal stem cells (eMSC) through the regulation of the expression of active WNT/β-catenin [22].

## 2. Exploring Notch Pathway Functions

Despite being an apparent simple pathway, Notch signaling has been implicated in several processes from early development to tissue homeostasis (Figure 2).

### 2.1. Regulating Cellular Dynamics, Developmental Patterning, and Reproductive System Homeostasis

Notch signaling exerts a pivotal influence over a spectrum of male and female reproductive events.

In Drosophila female germline, the ovaries exhibit distinct niches housing an average of 2–4 germline stem cells (GSCs), regulated by systemic signals and local factors. Notch signaling governs the formation and functionality of these ovarian germline niches, with its activation crucial for maintaining niche integrity and controlling GSC occupancy [23]. Overexpression of Mastermind regulates various Drosophila tissue niches, inhibiting the Hedgehog signaling pathway and reducing cadherin levels while elevating reactive oxygen species, consequently lowering GSC numbers—a characteristic of senescent GSC niches [24]. Notably, neither Notch activation in niche and cap cells nor GSC maintenance necessitates Notch ligands from GSCs themselves [25]. Yet, a comparable mechanism remains unidentified in the mammalian ovary. Furthermore, Notch and Delta also seem to be essential for the germline stem cell lineage in Drosophila testes [26], highlighting the crucial role of this pathway in reproductive processes.

In murine mouse models, it was observed that Notch receptors and ligands exhibit temporal expression within the testis, particularly within the seminiferous tubules across various cell types such as Sertoli cells, Leydig cells, and germ cells. This suggests the active involvement of Notch signaling throughout spermatogenesis, the intricate process culminating in the generation of mature sperm [27]. Similarly, in females, the Notch pathway plays a crucial role in orchestrating the development and function of the reproductive tract [28]. Meanwhile, within the female reproductive system, endometrium takes center stage, undergoing cyclic alteration driven by ovarian hormones and the embryonic signals ultimately fostering an environment conducive for embryo implantation and fetal growth. Moreover, during the preantral follicle phase, granulosa cells undergo development where they acquire receptors for follicle-stimulating hormones (FSHs), estrogen, and androgens, enabling them to react to these hormones appropriately. Additionally, several signaling pathways, including WNT2, WNT4, Notch2, Notch3, Jagged2, IHH, and DHH, are vital for the maintenance of granulosa cell function during their development [29]. Collectively, these observations emphasize the multifaceted role of Notch signaling in shaping and sustaining reproductive processes in both sexes.

### 2.2. Notch in the Orchestration of Cellular Proliferation, Differentiation, and Determinations of Cellular Fate

Pioneering studies in Drosophila have identified multiple roles of the Notch signaling pathway in regulating cell fate, cell proliferation, and cell death during development. These studies have demonstrated the pathway’s involvement in various developmental stages, initially reported for its neurogenic phenotype early in development, and later shown to play crucial roles in tissue homeostasis [12,14].

Accumulating evidence has demonstrated the conservation of these roles, as the ubiquitous presence of the Notch signaling pathway throughout the mammalian body underscores its fundamental role in dictating cellular behavior. Within the reproductive tract, this plays a critical role in orchestrating the development and function of both the female and male systems, influencing gametogenesis, hormone production, and organogenesis [28]. The human endometrium, a dynamic uterine lining, exhibits a cyclical pattern of proliferation, differentiation, and shedding throughout a woman’s reproductive years. [22]. Previous studies revealed that Notch signaling suppression reduces the proliferation rate of Notch1-positive LRSCs (label retaining stromal cells), leading to delayed endometrial repair in a mouse model simulating menstruation. Notch1 is required for proliferation and survival of EVT (extravillous trophoblasts) lineage [7,30]. 

Through direct regulation of the production of certain master regulatory transcription factors and distinguishing cytokines, the Notch pathway appears to regulate the differentiation of naïve CD4+ T cells into each of the Th fates. It is difficult to offer a plausible explanation for how various Notch ligands train naive CD4+ T cells to undergo such a wide collection of Th differentiation outcomes, given the shared machinery engaged by Notch interaction with each Notch ligand. Another perspective is that the Notch system controls T-cell growth during the priming phase and differentiated Th cells produce cytokines in response to subsequent stimulation instead of directly instructing Th destiny decisions [31]. The beginning of gene expression that results in cell fate is frequently followed by lateral inhibitory signaling, which causes the cells that are receiving the signal to revert to their initial fate [32].

### 2.3. Differential Effects of Notch Signaling on Cell Polarity

Cell polarity denotes the variations in shape, structure, and function that exist spatially within a cell. During organ development, Notch signaling controls cell–cell communication, cell polarity, and motility (Figure 2) [33]. Interestingly, it has been found that the availability and polarization of Notch ligands on the cell surface, as well as the shape of the cell, play important roles in refining Notch signaling output, with crucial implications for cell fate decisions during Drosophila development [34].

Furthermore, a recent study has unearthed a surprising twist in zebrafish neuroepithelial development. Mib1, an E3 ubiquitin ligase, appears to control apical–basal polarity, not through the canonical Notch pathway, but through its interaction with the FERM protein Ebp41l5. This versatile protein, known for its roles in cell polarity and EMT, seems to be the key player in Mib1’s orchestration of neuroepithelial architecture [34]. The disruption of cell polarity is typically seen as a late event in the development of tumors. Lgl1 has a crucial function in the maturation of the brain, as shown by Lgl1/embryos. A loss of cell polarity, lack of asymmetric Numb localization, and disruption of Notch signaling are all present in Lgl /neural progenitor cells [35]. The loss of the polarity gene was previously discovered to work in conjunction with Notch signaling to encourage the growth of tumors [36]. Epithelial tissue homeostasis and organization are both maintained through cell–cell adhesion and apicobasal polarity. It has been demonstrated that low-grade endometrial cancer and epithelial cells both exhibit decreased Notch signaling in response to apicobasal polarity disruption [37].

### 2.4. The Involvement of Notch in the Modulation of Cell–Cell Adhesion Processes

Notch signaling largely controls the activity of cadherin-based adherens junctions (AJ) and tight junctions (TJ) in the context of cell–cell adhesion. Evidence suggests that the cone cells in Drosophila eyes employ Notch signaling to direct nearby PPC (photoreceptor precursor cells) precursors to encircle them, while Notch oversees the restructuring process by regulating four adhesion genes in distinct ways. Notch triggers the expression of adhesion genes associated with the Nephrin group while inhibiting those linked to the Neph1 group [38]. Insights from another paper indicate that adhesion genes may not fully capture the role of Notch in recruiting PPCs. Notch is recognized for its diverse array of target genes, particularly involving various transcription factors crucial for determining PPC cell fate [39]. The intricate mechanisms of Notch-mediated cell adhesion hold exciting potential for therapeutic development. Manipulating this pathway could enhance stem cell engraftment and survival in regenerative medicine, regulate immune cell recruitment in inflammatory diseases, and even curb the spread of cancer cells through metastasis [40]. Figure 2 illustrates the complex role that Notch signaling plays in an array of vital functions within the cell, such as differentiation, proliferation, polarity maintenance, cell fate determination, epithelial–mesenchymal transition (EMT), and control of cell–cell adhesion.

### 2.5. The Notch Signaling Mechanism in the Regulation of EMT

Epithelial–mesenchymal transition (EMT) control has been linked to Notch signaling (Figure 2). Reduced invasive behavior in lung cancer cells and partial reversal of EMT are caused by the suppression of Notch1 [41]. EMT, which was first discovered in the context of embryonic development and involves the transition of epithelial cells to a mesenchymal cell phenotype [42], is essential for controlling invasion and metastasis [43]. The downregulation of Notch2 in NPC (nasopharyngeal carcinoma) cells elicited a phenotypic shift from epithelial to mesenchymal, characterized by suppression of an epithelial marker and upregulation of a mesenchymal marker. Conversely, Notch2 overexpression exerted opposing effects, further suggesting its negative regulatory role in NPC EMT [9]. Evidence suggests that numerous tumor microenvironmental stimuli, including hypoxia and estrogens, can cause epithelial mesenchymal transition (EMT), which is frequently activated by the Notch pathway. It has been demonstrated that the alternative estrogen receptor GPER (G protein- coupled estrogen receptor) causes EMT via Notch pathway activation [44].

### 2.6. Notch Signaling Guides Cellular Processes in Tissue Regeneration and Remodeling

By controlling cellular differentiation, proliferation, and cell-to-cell communication, Notch signaling is crucial for tissue regeneration and remodeling. In various types of tissues, including skin, liver, endometrium, and skeletal muscle, activation of the Notch pathway has been demonstrated to enhance tissue regeneration and repair. 

Recent discoveries have revealed Notch to be a key factor in liver regeneration and repair, as well as in liver metabolism, inflammation, and cancer [45]. Notch signaling is involved in the activation and differentiation of satellite cells, which are in charge of repairing and rebuilding skeletal muscle tissue following damage or injury, in the process of skeletal muscle regeneration [46]. The Notch pathway has a role in the differentiation and proliferation of epidermal stem cells and progenitor cells during skin regeneration. Epidermal stem cells are encouraged by Notch signaling to differentiate into intermediate keratinocytes before turning into mature epidermal cells [47].

### 2.7. The Contribution of Notch Signaling in Mediating the Adhesion of Blastocysts to the Human Endometrium

The juxtaposition of Notch1, DLL4, and JAG1 on the apical surface of the mid-secretory phase endometrium, alongside the presence of Notch receptors in the blastocyst’s trophectoderm, hints at a potential regulatory role for Notch signaling in establishing blastocyst–endometrial adhesion [21]. Emerging evidence suggests the blastocyst may trigger Notch signaling activation in the endometrium, potentially influencing endometrial receptivity. This notion is further supported by reduced or absent JAG1 immunostaining in the luminal epithelium of women with primary infertility during the mid-secretory phase [48]. Elucidating the molecular mechanism responsible for decidualization and implantation will help to improve the strategies to detect and prevent early pregnancy loss [49].

## 3. Unlocking the Notch: Exploring Its Impact on Reproductive Disorders and Beyond

Anticipated in its critical roles in various biological events, Notch has been implicated in a wide spectrum of human diseases. 

In this review, we emphasize the unspoken role of Notch in gynecological diseases because it facilitates comprehension of the underlying molecular process of those conditions. This knowledge is crucial for identifying key molecular players involved in developing and progressing diseases like endometriosis, adenomyosis, and polycystic ovary syndrome (PCOS). The Notch signaling elements show promise as potential diagnostic and prognostic markers for gynecological disorders. It can be anticipated that variations in the Notch pathway gene expression profiles can serve as prediction markers for the onset or progression of illnesses, offering clinicians significant tools for early identification and prognostic evaluation. The following details attempt to explore how Notch signaling may be involved in specific gynecological diseases (Figure 3):

### 3.1. Notch Signaling Implications in the Pathogenesis of Endometriosis

Amongst human tissues, the endometrium exhibits the singular characteristic of undergoing cyclical, dramatic remodeling to establish a receptive microenvironment for successful embryo implantation. Endometrium regeneration occurs after each menstrual cycle in humans and primates [21]. Different endometrial hormones, namely estrogen, progesterone, and human chorionic gonadotropin, are responsible for maintaining endometrial integrity. A significant portion (50–75%) of pregnancy loss can be attributed to implantation failure, where the blastocyst is unable to establish attachment with the maternal endometrium [49].

A chronic condition called endometriosis causes tissue like the inner lining of the uterus to grow outside it. This misplaced tissue reacts to hormones monthly, mimicking the menstrual cycle, but unlike the normal lining, it cannot be shed. This leads to pain and inflammation [56]. Normally, the menstrual cycle dating commences with the onset of menstruation (day 1). During this phase, termed the menstrual phase, declining estrogen levels induce endometrial shedding. The subsequent proliferative phase spans between menstruation and ovulation. Rising estrogen stimulates endometrial proliferation, characterized by thickening of the lining, with more active glands and lengthened spiral arteries [57]. Following ovulation, the secretory phase takes center stage, lasting until the next period. Early in this phase, progesterone levels rise, stimulating the secretion of glycogen and mucus within the endometrium. The mid-secretory phase witnesses a transformation, termed decidualization, that prepares the lining for a fertilized embryo. However, if pregnancy does not occur, the late secretory phase brings a hormonal plunge in both estrogen and progesterone. This hormonal shift triggers a constriction of the spiral arteries, leading to the breakdown of the endometrium, a process known as involution. The cycle then repeats [58].

A consortium of genes intricately collaborates to orchestrate the sophisticated hormonal and cellular processes inherent in the menstrual cycle. Notch4 appears to be more important for regulating cell growth (proliferation), while Notch1 seems to be more critical for cell specialization (differentiation). Disruptions in these processes may contribute to the development of various endometrial diseases, ranging from polyps to cancer [59]. During the mid-luteal phase (middle part of the menstrual cycle), the lining of the uterus (endometrium) increases the production of molecules called JAG1 and DLL4. As Notch ligands, they interact with Notch receptors on neighboring cells. This mechanism suggests that Notch signaling plays a role in making the uterus receptive to a fertilized egg. Women with endometriosis have lower levels of Notch signaling in their endometrial lining, a decrease linked to problems with a process called decidualization, which is essential for pregnancy [60]. Decidualization involves changes in the lining to support a growing embryo. Additionally, genes normally regulated by Notch1 show reduced activity in these women’s endometrial cells. It has been found that women with endometriosis have much lower levels of two molecules, JAG2 and DLL4, in their uterine lining compared to healthy women. These molecules act like signals for a cellular pathway, HEY1 and HES5, and were found to be less active in the endometrial tissue of these women. The levels of Notch1 and Notch4 expression were notably reduced in the endometrium of women with endometriosis compared to those of healthy women, while the expression of Notch2 and Notch3 remained unchanged. Thus, Notch signaling is involved in the regulation of several processes, including decidualization, implantation, uterine repair, and the periodic fluctuations of the menstrual cycle.

A large number of Notch receptors and ligands have been identified in the endometrium, placenta and blastocyst [21]. For the decidualization process to occur, Notch1 expression should be high, whereas there should be a downregulation of Notch1 for the stromal fibroblasts to differentiate into the decidual phenotype. This is very important for embryonic successful implantation and pregnancy [49]. Earlier studies have shown that induction of chorionic gonadotropin hormone can alter the morphology of the endometrium and the gene expression pattern in the uterus, including Notch1 [61]. The Notch inhibitor, Numb, acts as a Notch antagonist and can promote endocytic degradation of Notch [62]. During decidualization, Numb is significantly increased and there is a low expression of Notch1. There is an increase in Notch1 protein from proliferative to secretory phase of the uterine cycle. For a successful implantation, progesterone is essentially marked by the differentiation of the endometrium from proliferative to secretory phenotype. A deficiency in Notch1 signaling can impede progesterone-mediated decidualization in the endometrium, potentially contributing to gynecological pathologies [49]. Endometriosis is a condition where the cause remains unclear and there is no proper cure for it.

### 3.2. Notch1 and Its Contribution to the Genesis and Advancement of Adenomyosis

The non-neoplastic uterine disorder adenomyosis is distinguished by the ectopic presence and activity of endometrial glands and stroma within the myometrium. The symptoms include an enlarged uterus, painful menstrual cramps, abnormal bleeding, and infertility; this condition primarily affects women of reproductive age [63].

In adenomyosis, dysregulated Notch signaling promotes EMT, a cellular transformation crucial for its development. This involves decreased Numb expression, an antagonist of Notch, and concomitant upregulation of mesenchymal markers such as Snail, Slug, and N-cadherin [51]. In mouse models, it has been revealed that there is an increased levels of EMT markers and Notch 1 activation during the development of adenomyosis when compared to the control sample. However, it is very difficult to predict the onset of adenomyosis development in humans as it is only diagnosed once the patient presents symptoms [64]. Nevertheless, dysregulation of the Notch signaling pathway is implicated in the development and progression of adenomyosis [65]. Currently there are no tools available to predict whether the person will develop adenomyosis. Understanding the underlying molecular mechanisms might be crucial to diagnose and treat patients with adenomyosis at an earlier stage.

### 3.3. Notch Signaling: A Key Driver in the Pathogenesis of Asherman’s Syndrome

Asherman’s syndrome (AS) is a condition where scar tissue forms inside the uterus (intrauterine adhesions). This scarring can cause the walls of the uterus to stick together, leading to problems with menstruation, pain in the pelvis, difficulty becoming pregnant, repeated miscarriages, and issues with placental attachment during pregnancy [66]. Asherman’s syndrome (AS) is characterized by substantial impairment of the epithelial lining (epithelial compartment) and its differentiation processes due to disruptions in the Wnt and Notch signaling pathways. These disruptions significantly compromise endometrial functionality during the implantation window (WOI), a critical period for embryo attachment. Studies have revealed a reduction in the signaling interactions within the Notch pathway, specifically involving JAG1-Notch2, within the glandular, glandular secretory and luminal epithelial cells. This alteration correlates with the disrupted process of differentiation within the glandular epithelium in AS that can affect fertility and reproductive health [53].

### 3.4. Notch in the Ovary: Friend or Foe?

#### Deciphering Its Role in Follicular Health and Premature Ovarian Insufficiency

Primary ovarian insufficiency (POI) is characterized by diminished ovarian function ranging from irregular ovulation to complete cessation. It frequently results in premature menopause, which is defined as the cessation of ovulation before the age of 40 [67]. The incidence of POI has been steadily rising, characterized by significant heterogeneity and involvement of various genes affecting multiple biological processes such as hormonal signaling, metabolism, development, DNA replication, repair, and immune function. While a minority of patients show genetic links to known POI genes, a considerable portion remain undiagnosed genetically. Identifying the genetic basis of POI holds substantial benefits for patients and their families [68]. *Notch2* gene serves as a signaling factor that controls the formation of primordial follicles [69]. Various components of the Notch signaling pathway, including Notch2, Notch3, and Notch4, seem to play a role in the pathogenesis of POI within the OVO-Array cohort. This pathway, operating through contact-dependent signaling, is active during mammalian ovarian development and contributes to various processes such as follicle assembly, and the maturation, development, and initiation of meiosis [70].

### 3.5. Exploring Pre-Eclampsia’s Connection with Notch Signaling

Preeclampsia/eclampsia, a complex hypertensive condition occurring during pregnancy, continues to be a notable contributor to maternal and perinatal health complications and deaths globally [71]. Marked by the sudden onset of high blood pressure and organ impairment, notably impacting the kidneys and liver, managing preeclampsia/eclampsia presents significant obstacles in obstetric medicine [72]. The expression levels of Notch1 and Notch4 exhibit a significant reduction in preeclamptic placental tissue when compared to placental tissue from normotensive pregnancies. This observation implies that compromised Notch signaling pathways might play a contributory role in the pathogenesis of preeclampsia [52]. Research also revealed that the lack of JAG1 expression was frequently observed in perivascular and endovascular cytotrophoblasts in preeclampsia [73]. 

### 3.6. The Involvement of Notch Signaling in the Etiology of Polycystic Ovarian Syndrome

PCOS presents as a diverse endocrine syndrome characterized by the interplay of dysfunctional ovarian function and elevated androgen levels, with careful exclusion of other diagnoses like hyperprolactinemia and non-classical congenital adrenal hyperplasia being essential for accurate diagnosis [74]. Low expression of Notch1 and overexpression of Notch3 can have deleterious effects on uterine receptivity in the case of PCOS patients. An increase in Notch3 expression can be related to the exalted endometrial thickness and hyperplasia that are prominent in women with PCOS [50].

An area of research that needs more attention in the context of reproduction is the regulation of Notch signaling. Considering the multifaceted roles of this signaling pathway in the reproductive tract, from its development and normal function to its involvement in disease states, intensified research efforts are vital [28].

### 3.7. Unraveling the Role of Notch Signaling Pathway in Infertility: Insights and Perspectives

Infertility represents a pathological condition of the male or female reproductive system characterized by the inability to achieve conception following a period of at least 12 months of consistent unprotected sexual activity. The Notch signaling pathway serves pivotal functions in both the developmental processes and physiological operations of the male and female reproductive systems. Within the intricate milieu of the female reproductive tract, the Notch signaling pathway operates under the tight control of hormonal regulation, intricately modulating pivotal reproductive events crucial for the optimal function of both the ovaries and the uterus [28]. Irregular modulation of the Notch signaling pathway or genetic mutations can induce excessive or insufficient activation of its receptors, consequently leading to either upregulation or downregulation of Notch signaling. Such disturbances have the potential to interfere with the typical physiological processes within the ovary, potentially culminating in infertility [75]. Histological analysis reveals that endometria of infertile women often display reduced levels of Notch1 and DLL1 expression, along with elevated levels of Numb expression [54]. Multiple clinical investigations have identified mutations in Notch2 within the ovarian tissues of individuals diagnosed with primary ovarian insufficiency (POI) [76,77]. Recent investigations have revealed the emergence of lateral communication within the Notch signaling pathway across diverse cellular populations within ovarian tissues. Specifically, research has demonstrated that Jag1 expressed in oocytes triggers the activation of Notch2 or Notch3 receptors in granulosa cells, highlighting intercellular communication facilitated by Notch signaling. These studies provide evidence supporting the critical role of the Notch signaling pathway in preserving ovarian fertility through developmental regulation and modulation of granulosa cell function [55].

## 4. Role of Notch in Other Diseases: Cancer, Cardiovascular Diseases and Neurodegenerative Disorders

### 4.1. Cancer

Aberrant Notch signaling is seen in many cancers like T-cell acute lymphoblastic leukemia, acute myeloid leukemia, and cancers of the cervix, breast, colon, pancreas, skin, and brain. Not only is Notch required for cellular transformation, but it also facilitates crosstalk between transformed cells and supporting tissues during tumorigenesis [78]. A new tentative approach for cancer therapy is the inhibition of Notch signaling, which causes growth arrest and differentiation in cells with activated Notch pathways [3]. Depending on the tumor origin, Notch can influence differentiation, cellular metabolism, and cell cycle progression in a variety of ways that either promote or inhibit tumor growth [79]. Genome flaws, including both genetic and epigenetic changes, can modify Notch signaling outputs and promote carcinogenesis. The field of epigenetics studies heritable modifications of gene activity that take place without a change in the DNA sequence [3].

### 4.2. Cardiovascular Diseases

Cardiogenesis is a multistage, overlapping developmental process that involves cell proliferation, differentiation, and morphogenesis [80]. Genetic and in vitro research has linked Notch signaling to the growth and upkeep of the cardiovascular system by directly influencing the biological processes of cardiomyocytes and vascular cells (endothelial and vascular smooth muscle cells) [4]. Diseases affecting the heart’s vasculature have also been linked to abnormal Notch signaling [81]. The dependence of human embryogenesis on functional Notch signaling underscores its pivotal role in shaping the heart. The key involvement of Notch signaling during human heart development was well demonstrated, as a reduction in the transcriptional activity or haploinsufficiency of the downstream genes can result in congenital heart disease [82,83] or in developmental syndromes affecting the heart [83,84]. Similarly, understanding the role of Notch in vascular development through Drosophila research has provided insights into CADASIL, a rare cerebrovascular disease caused by *NOTCH3* gene mutations [12].

### 4.3. Neurodegenerative Disorders

Neurodegenerative diseases have previously been associated with many Notch gene mutations and proteins [85]. Notch signaling has been demonstrated to mediate the brain toxicity of a number of environmental variables and to play a modulatory role in a number of neurodegenerative disease model animals [5]. The Wnt pathway which is involved in cell fate determination, survival, and proliferation, is one of the many significant pathways known to interact with the Notch pathway [86]. In aging and age-related diseases, the interaction and balance between these various pathways may be disrupted [85]. Several other studies have demonstrated that Notch signaling is abnormally upregulated after ischemia damage [87,88]. 

### 4.4. Alagille Syndrome

Beyond cancer and neurodegeneration, Drosophila models have illuminated developmental defects linked to Notch dysregulation. Alagille syndrome, characterized by severe liver and kidney malformations, has been linked to mutations in the *JAG1* gene, a crucial Notch ligand. Drosophila models have revealed how these mutations disrupt Notch signaling in vital embryonic tissues, contributing to the characteristic features of the syndrome [89].

## 5. Prospective Therapeutic Targets Associated with the Control of Notch Signaling

The growing knowledge provided by different models regarding the functions and mechanisms of Notch signaling provides invaluable insight into pathological mechanisms and contributes to the potential therapeutic strategies targeting Notch. Targeting the Notch pathway means developing drugs or therapies that specifically affect this pathway to treat certain diseases including specific reproductive disorders discussed above. Below, we discuss current advances in Notch-targeting drugs and their potential applications.

### 5.1. γ-Secretase Inhibitors

The Notch signaling cascade culminates in γ-secretase-dependent intramembranous proteolysis of the Notch receptor. This liberates the Notch intracellular domain (NICD), which translocates to the nucleus and functions as a coactivator, driving the expression of downstream target genes. Consequently, therapeutic efforts have focused on γ-secretase inhibitors to impede Notch signaling and potentially treat associated diseases. DAPT(N-[N-(3,5-difluorophenacetyl)-l-alanyl]-s-phenylglycinetbutylester), a compound commonly utilized as a specific inhibitor of γ-secretase, acts by impeding the cleavage process of γ-secretase at the S3 site of the Notch receptor. This inhibition prevents the generation of the soluble Notch intracellular domain (NICD) protein. Consequently, it disrupts the association of the NICD with CSL (CBF1, Su(H), and Lag-1 proteins) and the co-activator Mam in the nucleus, thereby inhibiting the activation of target gene transcription [90]. However, a major challenge lies in the enzyme’s pleiotropic substrate specificity. γ-Secretase also cleaves other transmembrane proteins, notably the amyloid precursor protein (APP), and its inhibition can lead to unwanted accumulation of APP cleavage products, potentially contributing to off-target effects and hindering the therapeutic potential of γ-secretase inhibitors [91].

### 5.2. Antibodies Targeting Notch Receptors

Over the past few decades, there has been a surge in the development of Notch signaling inhibitors, including targeted small molecules and blocking antibodies. These agents are being actively investigated in preclinical and clinical settings for their therapeutic potential in both solid and hematological malignancies [92,93]. By complementing the development of small molecule Notch inhibitors, researchers have also generated monoclonal antibodies targeting Notch signaling components. Notably, MEDI0639, an antibody specific to DLL4, has been shown to inhibit DLL4 binding to Notch1. Interestingly, this antibody promotes human umbilical vein endothelial cell (HUVEC) growth and human vessel formation [94]. These antibodies may bind to Notch receptors with specificity and block ligand binding, which prevents receptor activation. Due to the intricacy of the Notch signaling system and potential off-target effects, the efficacy of these antibodies in clinical studies has been constrained [95]. Numerous small molecule inhibitors designed to modulate the Notch pathway such as siRNA, and monoclonal antibodies targeting both Notch receptors and ligands, have undergone development and are presently undergoing clinical trials. Moreover, the efficacy of Notch inhibitors has been investigated in conjunction with conventional cytostatic agents or targeted therapeutics within the context of phase I or phase II assessments [96]. The anti-Notch1 monoclonal antibody, OMP-52M51 (brontictuzumab), has undergone clinical evaluation for its efficacy in various malignancies, including lymphoid malignancies [97], solid tumors [98], adenoid cystic carcinoma (ACC) [99], and metastatic colorectal cancer (CRC). Notch2 has been associated with numerous cancer types, with anti-NRR2 emerging as a potential treatment option for melanomas and certain B-cell leukemias. These conditions are correlated with alterations in the Notch2 gene, such as amplification, overexpression, or mutations [94].

The diversity of Notch receptors is one of the biggest challenges in the development of antibodies that target Notch receptors. Mammals have four Notch receptors, each with distinct ligand-binding properties and expression profiles. Therefore, it is difficult to create antibodies that can target a single Notch receptor without impacting others.

### 5.3. Ligand-Mimicking Molecules 

Using ligand-mimicking compounds that may selectively activate or inhibit particular Notch receptors could be another method of regulating the Notch pathway. These compounds have the ability to control Notch signaling and imitate the binding of natural ligands. The structural intricacy of the Notch receptors and their ligands makes the production of these compounds difficult. Other proteins have the potential to function as Notch ligands and activate Notch through a different pathway. This pathway involves additional transmembrane proteins like Dner, F3/contactin-1, and NB-3/contactin-6. It is important to note that these Notch ligands bind to Notch receptors with lower affinity compared to the usual Notch ligands, mainly because they lack a specific structural region known as Delta/Serrate/Lag-2 (DSL) [100]. 

In preclinical cancer models, it has been demonstrated that DLL4-Fc, a recombinant protein that mimics the binding of Delta-like ligand 4 (DLL4), activates Notch signaling and inhibits tumor development [101]. In a study published in 2006, the role of DLL4 in tumor angiogenesis was investigated using preclinical models. It was found that DLL4 was highly expressed in tumor vasculature and blocking DLL4 signaling with a soluble form of DLL4 inhibited tumor angiogenesis and growth. These results highlighted the potential therapeutic implications of targeting DLL4 in cancer treatment [102]. Another study provided additional insights into how DLL4 functions in controlling angiogenesis. The study revealed that DLL4 serves as a suppressor of angiogenesis by influencing the behavior of endothelial cells via Notch signaling. Inhibiting DLL4-mediated Notch activation using soluble DLL4 or DLL4-Fc led to heightened angiogenesis and the formation of abnormal blood vessels, underscoring the significance of DLL4–Notch signaling in regulating angiogenesis [103].

## 6. Conclusions

In conclusion, Notch signaling emerges as a master orchestrator, exquisitely wielding its influence over a vast spectrum of physiological and pathological processes, including intricate aspects of human reproduction and diverse disease manifestations. Its multifaceted role in the reproductive system, encompassing endometrial cancer, regeneration, and endometriosis, underscores its potential as a therapeutic target. Beyond the realm of reproduction, Notch’s intricate web extends to the etiology of various diseases, including cancer, cardiovascular ailments, and neurodegenerative disorders, solidifying its profound impact on human health. 

Drosophila, with its remarkably conserved Notch pathway and rapid genetic tractability, becomes a potent investigative tool, enabling in vivo dissection of human disease mechanisms. This empowers researchers to unravel the intricate downstream consequences of Notch dysregulation, paving the way for the development of targeted therapeutic interventions across a diverse landscape of pathologies, from neurodegeneration and cancer to developmental malformations. As our understanding of the molecular underpinnings of Notch signaling and its intricate interplay with other pathways deepens, the discovery of innovative treatment strategies for a multitude of disorders becomes increasingly attainable.

Strategies including γ-secretase inhibitors, antibodies targeting Notch receptors, and compounds that mimic ligands are being investigated as potential treatment targets. However, obstacles such as off-target effects and the variety of Notch receptors highlight the need for more studies to improve and tailor these treatment modalities.

Notch signaling research is still being conducted, and it not only advances our understanding of reproductive physiology but also holds promise for creating targeted treatments for various illnesses, including cancer, neurological diseases, and reproductive problems. The increasing understanding of the complexities of Notch signaling provides opportunities for novel therapeutic approaches that may improve patient outcomes and further the field of reproductive medicine.

## Figures and Tables

**Figure 1 ijms-25-05423-f001:**
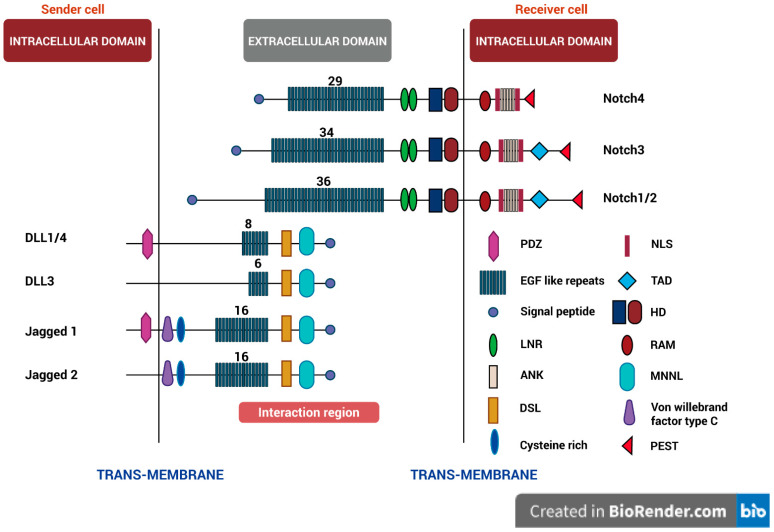
Ligands and Receptors in the Mammalian Notch Signaling Pathway. It illustrates the intricate cell–cell communication within the Notch signaling pathway. A ligand-expressing cell (Sender) engages with the Notch receptor situated on the neighboring cell (Receiver), thereby instigating a signaling cascade that orchestrates diverse cellular responses and regulatory mechanisms. It delves into the unique compositions of Notch signaling components in different animals. Notch receptors (Notch1–4) serve as the target of interaction between numerous types of Notch ligands, such as Delta-like (DLL1, DLL3, DLL4) and Jagged (JAG1, JAG2) to initiate signaling pathways involved in different cellular activities. Each domain is represented in the right corner as follows: PDZ domain, EGF like repeats, Signal peptide, LNR (Lin-12-Notch Repeats), ANK (Ankyrin), DSL (Delta/Serrate/Lag-2), Cysteine-rich, NLS (Nuclear localization signal), TAD (Transactivation domain), HD (Heterodimer), RAM (RBPJk-associated module), MNNL (membrane-bound Notch N-terminal Fragment L), Von Willebrand factor type C, and PEST (sequence rich in proline (P), glutamine (E), serine (S), and threonine (T)).

**Figure 2 ijms-25-05423-f002:**
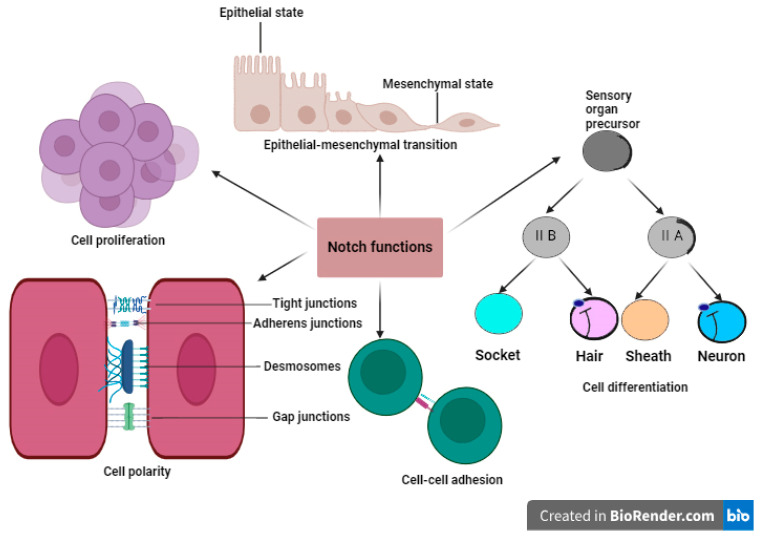
Multifaceted role of Notch signaling during development. This figure shows the multifaceted role of Notch signaling in diverse cellular processes such as epithelial–mesenchymal transition (EMT), a process where stationary epithelial cells transform into motile mesenchymal cells, important for sculpting organs. Additionally, Notch signaling can have proliferative effects, promoting cell division in specific contexts during development. This is crucial for proper tissue growth and expansion. Furthermore, Notch signaling plays a key role in maintaining cell polarity, the distinct organization of a cell with top and bottom sides, crucial for proper tissue formation. The differentiation process is illustrated by Drosophila sensory organ precursor that serves as a model system, ultimately generating four specialized cell types: hair, socket, sheath, and neuron. Finally, Notch signaling contributes to cell–cell adhesion, ensuring proper tissue architecture. By orchestrating these diverse functions, Notch signaling acts as a master conductor during development.

**Figure 3 ijms-25-05423-f003:**
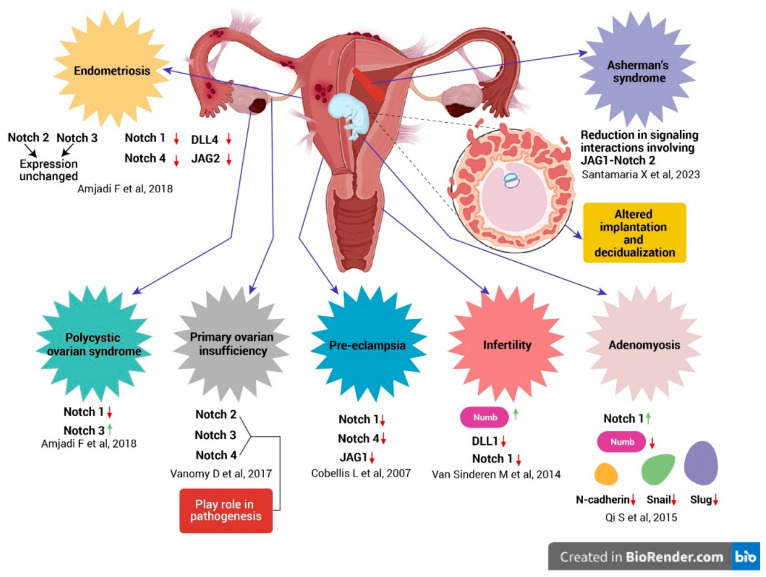
Notch receptor–ligand expression anomalies in gynecological pathologies: Insights and Implications. Notch receptors and ligands exhibit aberrant regulation in several pathological gynecological conditions like endometriosis [50], adenomyosis [51], pre-eclampsia [52], PCOS (Polycystic ovarian syndrome) [50], Asherman’s syndrome [53], infertility [54], and POI (primary ovarian insufficiency) [55] relative to healthy controls. It also unveils the captivating influence of Notch signaling in shaping the intricate processes of implantation and decidualization during pregnancy.

## Data Availability

Not applicable.

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
