# Peer review of "Notch Signaling: An Emerging Paradigm in the Pathogenesis of Reproductive Disorders and Diverse Pathological Conditions"

_ijms, 2024, doi:10.3390/ijms25105423_

Round 1

Reviewer 1 Report

Comments and Suggestions for Authors

The manuscript explores a captivating area of research related to Notch signaling and its potential role in reproductive health. However, there are several areas that require attention to improve the overall clarity and impact of the review.

Major Comments:

1.     Address the redundancy issue in the manuscript. Many lines show that they are directly taken from internet sources. For example, in lines 117-118, where the definition of endometriosis appears to be directly taken from the World Health Organization (WHO) description. It is crucial to present information in a manner that reflects originality.

2.     The review lacks organization and flow. for example,  Lines 122-133 show difficulty in comprehension. The paragraph could benefit from a different approach. Consider providing an overview of the cyclical changes in the menstrual cycle, followed by a discussion of the genes involved, details on decidualization, implantation, and the regulation of these processes by Notch signaling.

3.     Given the extensive discussion of various genes, it is advisable to incorporate a diagram illustrating the correlation between different gynecological pathologies and the associated Notch receptors and ligands.

4.     The information on infertility is currently weak and lacks clarity. Elaborate on the potential role of Notch signaling in infertility and suggest how it might be addressed. Strengthen the connection between Notch signaling and infertility to provide a more comprehensive perspective.

5.     Figure 2: Figure 2 is lacking in strength and informativeness. Consider revising or supplementing Figure 2 to enhance its clarity and communicative value.

6.     Writing font used in the manuscript is inconsistent; Check for variations in how author names, publication years, and titles are presented in the references. Correct any discrepancies to maintain a uniform appearance.

Comments on the Quality of English Language

Quality of English language is fine however some sentences are difficult to follow.

Reviewer 2 Report

Comments and Suggestions for Authors

Notch signalling pathway is crucial for many diseases including malignant tumors and reproductive failure-related lesions so the topic of this review is dramatically important. At the same time the authors describe only adenomiosis, endometriosis and PCOS while other conditions such as Asherman’s syndrome and ovarian insufficiency were not mentioned. Although there are some investigations related to these diseases with Notch signalling role as a central part of the pathogensis (Santamaria, X., Roson, B., Perez-Moraga, R. et al. Decoding the endometrial niche of WAsherman’s Syndrome at single-cell resolution. Nat Commun 14, 5890 (2023) and ovarian insufficiency (Moldovan GE, Miele L, Fazleabas AT. Notch signaling in reproduction. Trends Endocrinol Metab. 2021 Dec;32(12)). It would be better to add comprehensive review about more reproductive disorders. Although the part about target drugs should be expanded and key clinical trials should be discussed more comprehensively. 

Round 2

Reviewer 1 Report

Comments and Suggestions for Authors

The manuscript has been revised with a clear description of figures and text.

The manuscript reads well now.